# L-Arginine Ameliorates Defective Autophagy in GM2 Gangliosidoses by mTOR Modulation

**DOI:** 10.3390/cells10113122

**Published:** 2021-11-11

**Authors:** Beatriz Castejón-Vega, Alejandro Rubio, Antonio J. Pérez-Pulido, José L. Quiles, Jon D. Lane, Beatriz Fernández-Domínguez, María Begoña Cachón-González, Carmen Martín-Ruiz, Alberto Sanz, Timothy M. Cox, Elísabet Alcocer-Gómez, Mario D. Cordero

**Affiliations:** 1Research Laboratory, Oral Medicine Department, University of Sevilla, 41009 Sevilla, Spain; Beatriz.CastejonVega@glasgow.ac.uk; 2Centro Andaluz de Biologia del Desarrollo (CABD, UPO-CSIC-JA), Facultad de Ciencias Experimentales (Área de Genética), Universidad Pablo de Olavide, 41013 Sevilla, Spain; arubval@upo.es (A.R.); ajpulido@upo.es (A.J.P.-P.); 3Department of Physiology, Institute of Nutrition and Food Technology “José Mataix Verdú”, Biomedical Research Center, University of Granada, 18071 Granada, Spain; jlquiles@ugr.es; 4Cell Biology Laboratories, School of Biochemistry, University of Bristol, Bristol BS8 1TD, UK; Jon.Lane@bristol.ac.uk; 5Acción y Cura Para Tay-Sachs (ACTAYS), 28220 Madrid, Spain; beatriz@actays.org; 6Department of Medicine, University of Cambridge, Cambridge CB2 0QQ, UK; mcb23@medschl.cam.ac.uk (M.B.C.-G.); tmc12@medschl.cam.ac.uk (T.M.C.); 7Biosciences Institute, Newcastle University, Newcastle upon Tyne NE4 5 PL, UK; carmen.martin-ruiz@newcastle.ac.uk; 8Institute of Molecular, Cell and Systems Biology, University of Glasgow, Glasgow G12 8QQ, UK; Alberto.SanzMontero@glasgow.ac.uk; 9Departamento de Psicología Experimental, Facultad de Psicología, Universidad de Sevilla, 41009 Seville, Spain; ealcocer1@us.es; 10Instituto de Investigación e Innovación en Ciencias Biomédicas de Cádiz (INiBICA), 11009 Cadiz, Spain; 11CIBER de Enfermedades Respiratorias (CIBERES), Instituto de Salud Carlos III, 28220 Madrid, Spain

**Keywords:** autophagy, mTOR, GM2 gangliosidosis, L-arginine

## Abstract

Aims: Tay–Sachs and Sandhoff diseases (GM2 gangliosidosis) are autosomal recessive disorders of lysosomal function that cause progressive neurodegeneration in infants and young children. Impaired hydrolysis catalysed by β-hexosaminidase A (HexA) leads to the accumulation of GM2 ganglioside in neuronal lysosomes. Despite the storage phenotype, the role of autophagy and its regulation by mTOR has yet to be explored in the neuropathogenesis. Accordingly, we investigated the effects on autophagy and lysosomal integrity using skin fibroblasts obtained from patients with Tay–Sachs and Sandhoff diseases. Results: Pathological autophagosomes with impaired autophagic flux, an abnormality confirmed by electron microscopy and biochemical studies revealing the accelerated release of mature cathepsins and HexA into the cytosol, indicating increased lysosomal permeability. GM2 fibroblasts showed diminished mTOR signalling with reduced basal mTOR activity. Accordingly, provision of a positive nutrient signal by L-arginine supplementation partially restored mTOR activity and ameliorated the cytopathological abnormalities. Innovation: Our data provide a novel molecular mechanism underlying GM2 gangliosidosis. Impaired autophagy caused by insufficient lysosomal function might represent a new therapeutic target for these diseases. Conclusions: We contend that the expression of autophagy/lysosome/mTOR-associated molecules may prove useful peripheral biomarkers for facile monitoring of treatment of GM2 gangliosidosis and neurodegenerative disorders that affect the lysosomal function and disrupt autophagy.

## 1. Introduction

Autophagy is a degradation and clearance function of the lysosome that is critical for cellular homeostasis [1]. Where the digestive function is defective, as in the inherited Lysosomal Storage Disorders (LSD), accumulation of undegraded substrates in the lysosomal compartment can impair this fusion process [2]. As a result, in most of these genetic diseases, autophagic flux is arrested-with the consequential accumulation of other autophagy substrates including cell debris and organelles such as mitochondria, as well as the cargo protein, sequestosome-1-also known as the ubiquitin-binding protein p62 (SQSTM1/p62) [1,2].

Autophagy is regulated by diverse mechanisms, each of which serves as a potential axis for therapeutic intervention. The mechanistic target of rapamycin (mTOR) is a highly conserved serine/threonine kinase that serves as a master regulator of metabolic processes centred on autophagic control [3]. Moreover, the action of mTOR promotes lysosomal biogenesis and sustains the functional activity and integrity of this cell compartment [4]. Modulation of mTOR activity has been reported in Pompe disease, the first LSD to be biochemically characterised, where the significant muscular atrophy can be ameliorated by experimental induction of mTOR, which leads to substantial clearance of autophagic debris [5].

Here we have explored the role of autophagy in a class of lysosomal diseases-GM2 gangliosidosis Tay–Sachs and Sandhoff diseases-which principally affects sphingolipid recycling in the nervous system. The molecular cell pathology of these conditions reflects the striking accumulation of the primary substrate, GM2 ganglioside in neuronal lysosomes. That is due specifically to impaired hydrolysis in situ by β-hexosaminidase A (HexA). HexA is assembled as a functional heterodimer of α and β subunits. In humans, these proteins are encoded by *HEXA* and *HEXB* respectively, and the cognate subtypes are known as the genetically distinct, Tay–Sachs (Online Mendelian Inheritance in Man, OMIM, #272800) or Sandhoff (OMIM # 268800) Diseases [6,7]. Since the neurodegenerative features are clinically indistinguishable, diagnosis relies primarily on enzymatic assays selective for the different β-hexosaminidase isozymes A (α/β heterodimer; HexA); B (β/β homodimer; HexB) and S (α/α homodimer; HexS).

According to our study, we report that, when compared with healthy control cells, there is an enhanced expression of p62/SQSTM1 and build-up of autophagosomes, indicating impaired autophagic flux. This abnormality in fibroblasts from patients with GM2 gangliosidosis was confirmed by biochemical analysis and electron microscopy. Further, we observed an excess release of mature cathepsin B isoform and hexosaminidase A into the cytosol, which shows that lysosomal permeability is pathologically increased. GM2 fibroblasts had inappropriately diminished mTOR signalling with reduced basal mTOR activity. It is noteworthy that L-arginine supplementation of the diseased cells partially ameliorated these cytopathological abnormalities and immediately suggests an avenue for facile therapeutic exploration in this cruel disease.

## 2. Material and Methods

### 2.1. Ethical Statements

The work described was approved by the Ethics Committee of the Virgen de la Macarena and Virgen del Rocio University Hospitals, Seville, Spain, according to the principles of the Declaration of Helsinki and International Conferences on Harmonization and Good Clinical Practice Guidelines with the code 0795-N-15. All the participants in the study or their legal representatives gave written informed consent before the start of this study. Patients and samples were selected and isolated respectively from Hospital Clínico Santiago de Compostela (Dr Sánchez Pintos), Hospital 12 de Octubre (Dr Morales Conejo), Hospital La Fe de Valencia (Dr Pitarch Castellano), Hospital San Pedro de Logroño (Dr García Oguiza), Hospital Niño Jesús de Madrid (Dr González Gutiérrez-Solana), Hospital Son Espases (Dr Inés Roncero). The suspected diagnosis of GM2 gangliosidosis had been made by specific tissue (skin biopsies) enzyme studies at the appropriate specialist reference centre (Centre de Diagnòstic Biomèdic, Hospital Clínic de Barcelona), and Tay–Sachs or Sandhoff diseases were confirmed by molecular analysis of the *HEXA* or *HEXB* genes, respectively. The clinical presentation and diagnostic information are presented in the Appendix A.

For experimental protocol, electronic laboratory notebook was not used.

### 2.2. Fibroblast Culture

Fibroblasts from patients with GM2 gangliosidosis were obtained from skin biopsy samples according to the Helsinki Declarations of 1964, as revised in 2001 for this specific research project according to the approved ethical committee 0795-N-15. Control fibroblasts were commercial primary dermal fibroblast from Juvenile and Infant donors (Primacyt Cell Culture Technology GmbH, Schwerin, Germany). Two-line of control fibroblasts were used and represented by the mean of both compared with the different patients. Fibroblasts were cultured in high glucose DMEM (Dulbecco’s modified media) (Gibco, Invitrogen, Eugene, OR, USA). The medium was supplemented with 10% fetal bovine serum (FBS) (Gibco, Invitrogen, Eugene, OR, USA) and antibiotics (Sigma Chemical Co., St. Louis, MO, USA). Cells were incubated at 37 °C in a 5% CO_2_ atmosphere. The medium was changed every two days to avoid changes in pH.

### 2.3. Reagents

Trypsin and bafilomycin A1 were purchased from Sigma Chemical Co., (St. Louis, MO, USA). Anti-GAPDH monoclonal antibody from Calbiochem-Merck Chemicals Ltd. (Nottingham, UK). MnSOD, catalase and OGG-1 antibodies from Adipogen (San Diego, CA, USA). Phospho-mTOR, mTOR, phosphor-AKT, AKT, LC3, p62, LAMP-I, Cathepsin B, HexA and galectin-3 were obtained from Cell Signaling Technology. A cocktail of protease inhibitors (complete cocktail) was purchased from Boehringer Mannheim (Indianapolis, IN, USA). Grace’s insect medium was purchased from Gibco. The Immun Star HRP substrate kit was from Bio-Rad Laboratories Inc. (Hercules, CA, USA).

### 2.4. Structural Mutation Analysis

Human HexA and HexB cDNA sequences were downloaded from GenBank (Gene ID: 3073 and 3074) and used to interpret the functional effects of the mutations in this autosomal recessive disease with the web application Mutalyzed [8]. To investigate the effect of the mutations on protein folding, cognate 3D crystal structures were obtained from the PDB database (HEXA: 2GJX and HEXB: 1NOU) and viewed using PyMOL. The stability of the mutated structures was examined with the CUPSAT and SDM programmes; these have default values and enable torsion angles and free energy to be estimated, thus allowing changes in protein stability to be predicted.

### 2.5. Determination of β-Hexosaminidase Activities

Lysosomal hexosaminidase activities were measured as described previously [9]. Briefly, total β-hexosaminidase activity (A, B and S isozymes) was measured using the fluorogenic 4-methylumbelliferyl-2-acetamido-2-deoxy-D-glucopyranoside (4-MUG) substrate, whereas the HexA isozyme activity was determined with the sulphated, 4-methylumbelliferyl-7-[6-sulpho-2-acetamido2-deoxy-β-D-glucopyranoside] (4-MUGS) substrate, which is preferentially hydrolysed by the A and minor S isozymes.

### 2.6. Western Blotting

Whole cellular lysate from fibroblasts was prepared by gentle shaking with a buffer containing 0.9% NaCl, 20 mM Tris-HCl, pH 7.6, 0.1% Triton X-100, 1 mM phenylmethylsulfonylfluoride and 0.01% leupeptin. The protein content was determined by the Bradford method. Electrophoresis was carried out in a 10–15% acrylamide SDS/PAGE and proteins were transferred to Immobilon membranes (Amersham Pharmacia, Piscataway, NJ, USA). Next, membranes were washed with PBS, blocked over night at 4 °C and incubated with the respective primary antibody solution (1:1000). Membranes were then probed with their respective secondary antibody (1:2500). Immunolabeled proteins were detected by chemiluminescence method (Immun Star HRP substrate kit, Bio-Rad Laboratories Inc., Hercules, CA, USA). Western blot images were quantified using ImageJ software.

### 2.7. Electron Microscopy

Fibroblasts were fixed for 15 min in the culture plates with 1.5% glutaraldehyde in culture medium and then for 30 min in 1.5% glutaraldehyde-0.1 M NaCacodylate/HCl, pH 7.4. After three washes in 0.1 M sodium cacodylate/HCl, pH 7.4 for 10 min, the cells were post-fixed and stained with 1% aqueous osmium tetroxide, pH 7.4 for 30 min. After dehydration in increasing concentrations of ethanol, 5 min for each step: 50, 70, 90 and three cycles at 100% ethanol, the cells were impregnated and included in Epon, which was finally polymerised at 60 °C for 48 h. 60–80 nm sections were cut in a Leica ultracut S ultramicrotome (Leitz Microsystems, Wetzlar, Germany) and contrasted with uranyl acetate and lead citrate. Cell ultrastructure was examined by electron transmission microscopy (Zeiss LEO 906 E; Oberkochen, Germany).

### 2.8. Proliferation Rate

Two hundred thousand fibroblasts were cultured with or without the addition of L-arginine, L-leucine or DL-acetyl-leucine at different concentrations (1, 10, 5 mM) for 24, 48, 72, and 120 h. After discharging supernatant with dead cells, cells from three high-power fields were counted with an inverted microscope using a 40× objective

### 2.9. Cathepsin B and HexA Release

CatB and HexA redistribution from lysosomes/autophagolysosomes to the cytosol was assessed by immunofluorescence using antibodies against CatB or HexA and LAMP-I as a marker of lysosomal/autophagolysosomal compartment. Fibroblasts were grown on 1 mm with glass coverslips for 48 h in normal growth medium. Cells were rinsed twice with PBS and fixed using 3.8% paraformaldehyde for 5 min at room temperature, and permeabilised with 0.1% saponin for 5 min. Then, glass coverslips were incubated at 37 °C with primary antibodies diluted as appropiated (1:100–1:500) in PBT (PBS with 0.1% Tween 20) for an 1 h at 37 °C and rinsed twice with PBT. The secondary antibody, diluted 1:200 in PBT, was incubated for 45 min at 37 °C. The coverslips were then rinsed twice with PBT and mounted onto microscope slides using Vectashield Mounting Medium with DAPI and analysed using an upright fluorescence microscope (Leica DMRE, Leica Microsystems GmbH, Wetzlar, Germany). Images were taken using a DeltaVision system (Applied Precision; Issaquah, WA, USA) with an Olympus IX-71 microscope using a 100× objective. In control fibroblasts, CatB-specific immunostaining shows punctate cytoplasmic structures surrounded by lysosomal/autophagolysosomal membrane proteins such as LAMP-I. After lysosomal permeabilisation, immunofluorescence of CatB or HexA reveals diffuse staining throughout the entire cell. Quantification of colocalised puncta were quantified using JACoP plugin in ImageJ.

### 2.10. Galectin 3 Puncta

Lysosomal permeabilisation was also evaluated by quantification of intracellular galectin puncta, as previously described.

### 2.11. Protein Synthesis

Briefly, the cells were incubated with serum-free DMEM for 90 min followed by incubation with 1 µM puromycin (an analogue of tyrosyl-tRNA; Invitrogen; A11138-03), for 30 min. The amount of puromycin incorporated into nascent peptides was then evaluated by Western blot using an antibody to puromycin.

### 2.12. L-Arginine Treatment in Patients

Two patients with GM2 gangliosidosis (Juvenile Tay–Sachs disease 3 and Sandhoff disease 2, (see Appendix A), who had not been exposed to any drug or vitamin/nutritional supplements, were supplemented with oral L-arginine (Nutricia) for 8 months (0.3 g/Kg/day). After 8 months of treatment, heparinised blood samples were collected 24 h after the last dose. No significant changes in routine clinical laboratory tests conducted on blood serum or plasma, including measures of renal and hepatic function, were observed (data not shown).

### 2.13. Statistical Analysis

Data in the figures is shown as mean ± SD. Data between different groups were analysed statistically by using ANOVA on Ranks with Sigma Plot and Sigma Stat statistical software (SPSS for Windows, 19, 2010, SPSS Inc. Chicago, IL, USA). For cell-culture studies, Student’s t test was used for data analyses. A value of *p* < 0.05 was considered significant.

## 3. Results

### 3.1. Mutant Fibroblasts from Tay–Sachs Patients Showed Impaired Autophagic Flux

Six patients, 3 infantile and 3 juvenile, contributed to this study. The clinical characteristics of these patients are included in Appendix A. With the exception of juvenile patient 1 and infantile patient 3, the point mutations identified in the HEXA and HEXB genes were predicted to destabilise the intact protein. The latter, (allele 2), was especially noteworthy since the mutation was predicted to confer greater structural stability. However, the amino acid replacement directly affects the active site region. Patients 1 and 2 with the infantile-onset disease have non-coding mutations in close apposition to splice sites (Figure 1A–C and Appendix A). Those mutations were considered to be the most destabilising and to have a more profound effect on hexosaminidase A catalysis as revealed by the impaired activities (compared with the reference values from healthy individuals (Figure 1D)).

Here we explore the cellular pathogenesis of GM2 gangliosidosis. In culture, the skin fibroblasts obtained from patients had impaired growth rates and markedly abnormal lysosomal morphology (Figure 1E,F). Western blotting studies showed increased abundance of the immunoreactive autophagy markers, LC3-II and p62/SQSTM1; the intracellular accumulation of lysosomal substrates as determined by p62/SQSTM1 was confirmed by immunofluorescence confocal microscopy (Figure 1G,H).

To investigate the integrity of lysosomal maturation and the formation of autophagosomes, we used bafilomycin A1 as an inhibitor of the vacuolar H^+^ ATPase (vATPase). The bafilomycin A1 (BafA1) assay serves as a means to explore autophagosome/autophagolysosomal formation in the living cell. As expected, its effects on vATPase caused BafA1 to abrogate lysosomal acidification and intralysosomal digestion of substrates. In control cells addition of BafA1increased cellular abundance of LC3-II but the exposure of Tay–Sachs fibroblasts to the inhibitor did not affect baseline LC3-II staining, (Figure 1I). These findings strongly suggest that autophagosome processing is defective in these cells.

### 3.2. Autophagosome Accumulation with Arrested Autophagic Flux in Tay–Sachs Disease

Electron microscopy of fibroblasts obtained from patients with Tay–Sachs disease revealed an extensive accumulation of autophagosomes, an abnormality not present in control cells (Figure 2A,B). The pathological changes, with abundant multilamellar bodies, closely resemble those first reported in neurons obtained from the brains of infants with Tay–Sachs disease [10].

To distinguish between autophagosome–lysosome fusion or inefficient lysosomal degradation as factors in the accumulation of autophagosomes and altered autophagic flux, we examined lysosome–autophagosome fusion using tandem fluorescent-tagged LC3-II as an autophagosomal marker. We observed numerous yellow structures corresponding to autophagosomes in mCherry-GFP-LC3-II-expressing fibroblasts from Tay–Sachs patients, compared with those from control subjects (Figure 2C,E).

Engulfment of mitochondria by lysosomes and their digestion by mitophagy was explored by the use of high-resolution confocal microscopy to co-localise cytochrome C and LC3 (Figure 2D,F). Markedly increased abundance of engulfed, but incompletely digested mitochondria, confirmed that fusion of lysosomes with autophagosomes is impaired-an abnormality that was observed in fibroblasts from patients with Tay–Sachs disease, irrespective of their clinical severity.

### 3.3. Autophagosome Accumulation Is Associated with Increased Lysosomal Permeability

Degradation of autophagic cargo by acid hydrolases, including cathepsins, occurs in the autophagolysosomal compartment. Impaired autophagic flux with the accumulation of autophagosomes/autophagolysosomes can result from either reduced autophagosome–lysosome fusion or inefficient lysosomal degradation [1]. To distinguish between these mechanisms, we determined whether the impaired autophagic flux in cells affected by Tay–Sachs disease and deficient β-hexosaminidase A activity was associated with decreased activity of other lysosomal acid hydrolases. In this respect, the expression of CatB and CatD was increased in Tay–Sachs fibroblasts. (Figure 3A). To explore the possible role of CatB in the pathophysiology of Tay–Sachs disease, we examined the intracellular localisation of the protein by confocal immunofluorescence microscopy. In healthy fibroblasts, the CatB signal co-localised with the lysosomal membrane marker LAMP-1 (Pearson’s coefficient of correlation ~ Ctl: 0.89; Inf1: 0.21; Inf2: 0.14; Juv1: 0.078; Juv2: 0.19), indicating that, as expected, it is principally found in the lysosome/autophagolysosome compartment. However, in Tay–Sachs fibroblasts, the CatB immunofluorescence signal occurred diffusely throughout the cytosol and was only partially associated with the LAMP-I marker. These findings suggested that increased lysosome/autophagolysosome membrane permeabilisation is a pathological manifestation of the disease (Figure 3B,D). Of note, in this context, HexA immunostaining also revealed a diffuse pattern in the cytosol and a similarly reduced co-localisation with the lysosomal membrane marker, LAMP-I (Pearson’s coefficient of correlation ~ Ctl: 0.71; Inf1: 0.34; Inf2: 0.22; Juv1: 0.17; Juv2: 0.29) (Figure 3C,E). The latter observation was not only unexpected and dependent on the presence of residual immuno-reactivity of some mutant HexA proteins but also may reflect the consequences of protein aggregation.

Increased permeability of lysosomes in mutant fibroblasts was explored further by examining the appearance and distribution of galectin puncta in autophagolysosomes (Appendix A). These findings confirmed the enhanced lysosomal permeability of fibroblasts that we identified in Tay–Sachs disease. Analysis of isolated lysosomes obtained after cell fractionation confirmed the redistribution of HexA antigen with an increased fluorescence signal in the cytosol of Tay–Sachs fibroblasts compared with controls (Figure 3F); this altered distribution was reflected in a relative reduction of the lysosomal component. As depicted in Figure 3F, redistribution of mutant HexA occurred in fibroblasts in which increased cytosolic abundance of CatB was detected.

Given the organelle pathology that we observed in Tay–Sachs fibroblasts, we examined the subcellular distribution of the master regulator of autophagy and lysosomal biogenesis, Transcription Factor EB (TFEB). In healthy cells, TFEB is normally inactive and diffused in the cytosol in association with the surface of the lysosome, however, when the lysosomal function is inhibited, dephosphorylation of TFEB leads to its translocation to the nucleus where stimulates lysosomal biogenesis by actively upregulating the transcription of target genes harbouring the CLEAR element [11]. As predicted and illustrated in Appendix A, immunoreactive TFEB were abundant and concentrated in nuclei of fibroblasts obtained from patients with both subtypes of GM2 gangliosidosis.

### 3.4. Altered mTOR Pathway Is Associated with HexA Expression

TFEB is activated under conditions of restricted nutrition and energy generation, and in authentic models of diseases in which lysosomal clearance of intracellular debris is impaired. Given as above that we found subcellular localisation of TFEB from lysosomes to the nucleus in diseased fibroblasts, we investigated the potential engagement of the mechanistic target of rapamycin (mTOR), a kinase and primary regulator of autophagy and lysosomal biogenesis [3,4] in the TFEB translocation process. The down-regulated and phosphorylated forms of p-mTOR and p-AKT were decreased in fibroblasts from patients with Tay–Sachs compared with control cells (Figure 4).

### 3.5. Impaired mTOR/Autophagy and Lysosomal Membrane Permeabilisation (LMP) Are Also Associated with Sandhoff Disease

To test the common underlying cellular pathophysiology in GM2 gangliosidosis, we further explored the changes in Sandhoff disease, which is related to Tay–Sachs disease but due to mutations in HEXB with consequential effects on the β-subunit shared by the Hex A and Hex B isoenzymes. Fibroblasts obtained from two patients with Sandhoff disease with reduced enzymatic activities and confirmed HEXB mutations were studied (Appendix A). As with the point mutations detected in the patients with Tay–Sachs disease, those in Sandoff disease were predicted to destabilise the β-hexosaminidase structures (Appendix A). Similarly, autophagy was impaired in fibroblasts from these patients with increased LC3-II abundance and, accumulation of p62 and with p-mTOR species (Figure 5A). In addition, we observed increased expression of active CatB and CatD with enhanced release of CatB from the lysosome into the cytosolic compartment (Figure 5A–C) with abundant and concentrated TFEB in nuclei of fibroblasts obtained from patients with Sandhoff (Appendix A). Additional confirmation of these pathological effects was provided by the immunofluorescent detection of galectin puncta in autophagolysosomes (Appendix A), and identification of abundant multilamellar bodies and appearance of autophagosomes by electron microscopy (Figure 5B,D). To validate our findings in vivo, we used a mouse model lacking both Hex A and B activities as a result of targeted disruption of the hex β subunit gene (Sandhoff strain) which is an authentic model of acute human GM2 gangliosidosis (and Tay–Sachs disease). As in the different cellular models, the analysis of brain, spinal cord, brain stem and cerebellum was consistent with marked disruption of autophagy accompanied by inhibition of p-mTOR and increased expression of active cathepsin B, CatB (Figure 5E).

### 3.6. Transcriptomic Analysis Reveals Altered Molecular Pathways

To better define the molecular pathophysiology of cell injury in Tay–Sachs disease, a microarray expression profiling was carried out on fibroblasts cultured from control subjects and affected are available at http://www.ncbi.nlm.nih.gov/geo/ with code GSE184906 (accessed on 30 September 2021). Of the 135,750 transcripts examined, fibroblasts from patients with the severe infantile variants of the disease showed significant changes in 2141 transcripts when compared with control fibroblasts: 886 transcripts were upregulated and 1255 downregulated. Similar studies in fibroblasts from a patient with the more indolent juvenile variant revealed changes in the abundance of 1327 transcripts: 605 were upregulated, and 722 downregulated. Finally, fibroblasts affected by Sandhoff disease showed changes in the steady-state abundance of 1990 transcripts compared with those from control fibroblasts: 902 transcripts were upregulated, and 1088 downregulated. In-depth pathway analysis indicated changes in the expression of genes encoding proteins engaged in mTOR signalling, autophagy and other lysosomal processes (Appendix A and Supplementary Appendix A). Significant changes were observed in multiple genes implicated in these pathways. Of note, despite the variability observed between cells from different patients, several changes appear to be related to the pathobiological changes affecting lysosomal function that we report here. Expression of many genes of the mTOR pathway was downregulated, including ATP6V1C1 and Rictor. At the same time, Bcl2, an inhibitor of autophagy [12], and RAB7B, a negative regulator of autophagy flux [13], were upregulated. Arylsulphatase G (ARSG) and aspartylglucosaminidase (AGA), lysosomal enzymes, which are mutated in lysosomal diseases both in animals and humans, were also downregulated. The former changes are compatible with the finding of increased lysosomal permeability [2,14]. Similarly reduced abundance of the LAPTM4B protein has been linked to increased membrane permeability [15].

Of particular relevance, was the increased expression of the phosphoinositide-3-kinase, regulatory subunit 1 (PIK3R1) gene which encodes the regulatory domain (p85α) of the PI3K complex. This change was shared between both subtypes of GM2 gangliosidosis. Accordingly, we returned to the fibroblasts and confirmed the increased abundance of the PI3K protein in the fibroblasts from the patients (Appendix A). PI3K induces the phosphorylation of AKT and mTOR, but the phosphorylated forms of both were decreased; we propose that the enhanced expression of PI3K transcripts may reflect a compensatory change related to downregulation of the PI3K/AKT/mTOR pathway. To explore this phenomenon further, we investigated the phosphatase, PTEN, which canonically regulates the PI3K signalling cascade in a negative manner, thereby dampening downstream AKT/mTOR signalling [16]. According to our hypothesis, PTEN gene transcripts were overexpressed in fibroblasts from patients with GM2 gangliosidosis, compared with cells from control subjects (Appendix A).

### 3.7. Arginine Treatment Recovers mTOR Activity and Lysosomal Dysfunction

As a component of the master-regulator, mTOR complex 1 (mTORC1), mTOR links the availability of nutrients with cell growth and autophagy. Since mTORC1 activity is modulated by growth factors, stress, energy status and amino acids [17], and its function is altered in fibroblasts obtained from patients with Tay–Sachs and Sandhoff Diseases, we sought to determine whether it represents a potential therapeutic target.

Accordingly, we explored the effects of modulating mTOR activity by supplementing fibroblast cultures with amino acids. First, we evaluated the effect of the natural amino acids, L-arginine and L-leucine as well as acetyl-DL-leucine, a modified, cell-penetrant amino acid which has been shown to improve clinical symptoms in Niemann-Pick type C patients [18]. Under the experimental conditions used, L-arginine treatment significantly increased the rate of cell growth in fibroblasts obtained from patients with both GM2 gangliosidosis sub-types (Appendix A). It was notable that this improvement was not observed with either L-leucine (Appendix A) or acetyl-DL-leucine (Appendix A).

To understand the apparent specificity of the arginine effect *in vitro*, we conducted a more intensive analysis of the transcriptomic findings in GM2 gangliosidosis fibroblasts: this revealed changes in the expression of several genes that encode enzymes involved in arginine biosynthesis (Appendix AA–C and Appendix A). On account of this finding, we searched for indicators of altered arginine metabolism in patients with Tay–Sachs and Sandhoff disease. Concentrations in serum of L-arginine and nitric oxide (generated from arginine by the action of nitric oxide synthases) were moderately reduced in patients (Appendix AA–C). The most significant change in the transcriptomic analysis of arginine biosynthesis was argininosuccinate synthetase (ASS1), the third enzyme of the urea cycle which catalyses the formation of argininosuccinate from aspartate, citrulline, and ATP. Located on the outer mitochondrial membrane, ASS1 is the rate-limiting step for the formation of arginine *de novo* and a source of this substrate for nitric oxide synthases [19]. The transcriptomic findings in Tay–Sachs and Sandhoff were confirmed by reduced expression of ASS1 protein (Appendix AD–F). These findings raise the possibility that alterations in arginine biosynthesis contribute to the pathophysiology of these diseases.

On the other hand, the mTOR pathway was partially rescued in patients with Tay–Sachs disease after 120 h of exposure to L-arginine with a boost in the reduced protein synthesis of the patients (Figure 6A,B). These observations were also mirrored in fibroblasts from patients with Sandhoff disease (Appendix AA). Furthermore, we examined lysosome–autophagosome fusion. A significantly reduced number of yellow structures that correspond to autophagosomes was evident in mCherry-GFP-LC3-expressing Tay–Sachs fibroblasts after L-arginine treatment (Figure 6C). These results confirmed that mTOR activity and the fusion of autophagosomes and lysosomes could be improved by supplementation with L-arginine.

Given that altered lysosomal membrane permeability may be a critical factor in the pathophysiological alterations found in cells from patients with GM2 disease, we next evaluated the effects of L-arginine treatment on CatB release from lysosomes. Treatment with L-arginine induced a marked reduction in CatB expression levels in Tay–Sachs fibroblasts (Figure 7A), as well as an increased co-localisation of CatB signal with the LAMP-1 marker, suggesting the restitution of the permeability barrier (Figure 7A,C). A significant increase in HexA co-localisation with LAMP-1 was observed alongside the elevated expression of the mature form of HexA (Figure 7B,C)-findings shared between Tay–Sachs and Sandhoff diseases. (Appendix AB). To test whether the supplementation with L-arginine suppressed autophagy, by complementing the supply of amino acids, we examined the effects on TFEB localisation. After the addition of L-arginine, there was a notable decrease in the presence of nuclear TFEB an effect which would lead to a reduced transcriptional drive towards lysosomal biogenesis (Appendix A).

Finally, as an early exploration of proof of concept, two patients suffering from juvenile forms of Tay–Sachs and Sandhoff diseases TSD2 and Juvenile SD2) consented to take oral supplements of L-arginine (0.3 g/Kg/day) for 8 months. Although it was not possible to provide an objective evaluation of the neurological outcomes, family carers and physiotherapists reported improved coordination in both patients and suggested that their rate of cognitive deterioration was partially arrested. The effects of L-arginine supplementation were explored in mononuclear blood cells obtained from these patients. As shown in Figure 7D, administration of oral L-arginine partially restored mTOR expression. Moreover, the pathological abundance of cathepsin B was suppressed and accompanying this, the expression of HexA and arginosuccinate lyase protein in these cells was improved.

## 4. Discussion

Despite many initiatives, no treatment of proven safety and efficacy is available for patients stricken by any clinical subtype of GM2 gangliosidosis. While the genetic and biochemical basis for this disease have been well studied, an integrated description of its pathogenesis and the sequence of unitary steps that lead to its destructive neuroinflammatory effects is lacking. To explore the pathophysiological complexity of these sphingolipid diseases, we used living cells as a focus for a comprehensive molecular characterisation of their disordered cell biology. This platform facilitated corroborative investigations in the coherent model of these disorders in the genetically modified mouse, followed by early proof-of-concept studies carried out in two affected patients.

Here we describe metabolic derangements that accompany markedly impaired autophagy in human fibroblasts harbouring pathological defined HEXA and HEXB mutations. The diseased cells had poor growth rates in culture. Thus, they showed reduced ATP concentration and energy charge together with disrupted mitochondrial electron-transport and suffered oxidative stress. Despite the florid appearance of autophagic vacuoles, the accompanying increase of LC3-II protein might indicate either an enhanced autophagic drive or, as here, also with enhanced p62/SQSTM1 expression, impaired autophagic flux [20]. Blocked autophagic flux was confirmed by a BafA1 assay and by ultrastructural appearances. We noted, that defective autophagic degradation or interruption in autophagic flux has been shown in several lysosomal diseases such as Niemann-Pick disease type C, Gaucher and Pompe diseases [4,21,22,23].

Numerous abnormalities have been reported in association with lysosomal dysfunction. These include: changes in lysosomal enzymes, the volume and number of lysosomes and in membrane properties. All these abnormalities represent or drive the loss of functionality including autophagy [24]. For example, a reduction in members of the family of cysteine proteinases such as CatC and CatL, have been related to a compensatory transcriptional upregulation of CatB expression due to TFEB as well as autophagy dysfunction [25,26]. Furthermore, in Niemann-Pick disease C, the increased expression of mature protein forms for both CatB and CatD are associated with the build-up of autophagosomes [27].

Here we found that the deficiency of HexA was associated with a compensatory upregulation of mature CatB, CatD, and lysosomal permeabilisation. Furthermore, this permeabilisation was associated with the release of CatB and HexA. The pent-up machinery of autophagy and effects on intracellular metabolism are likely to have a strong bearing on the pathophysiology of this disease, given that florid end-stage changes are prominent neuropathological features that presage cell death in affected neurons distributed throughout the nervous system in the last phases of illness in patients with Tay–Sachs and Sandhoff diseases [24,27]. In its terminal phase, deranged autophagy is likely to contribute additionally to the pathological cascade by stimulating release of inflammatory cytokines through the agency of the p62/SQSTM1 signal [1]. As to the upstream drive to enhance autophagosome genesis in Tay–Sachs and Sandhoff diseases, we find evidence that this is due to nuclear translocation of TFEB, since it activates genes that orchestrate lysosomal biogenesis. The critical discovery of this fundamental process emerged from the brilliant realisation that coordinated lysosomal expression and regulation (CLEAR) represented a gene network which could be controlled by a single major transcriptional factor that recognised a key regulatory element common to effector proteins fundamentally implicated in lysosomal pathobiology [28]. The complex disturbance of lysosomal structure and function in fibroblasts from Tay–Sachs and Sandhoff patients as a consequence of reduced activity of the lysosomal β-hexosaminidases with increased autophagosome size can be attributed to TFEB activation as part of a response that will drive compensatory expansion of the lysosomal compartment. The enhanced TFEB nuclear localisation, we reported here in Tay–Sachs and Sandhoff fibroblasts is fully compatible with this process [28]. In this context we further explored the potential engagement of the multifunctional mTOR pathway as and found reduced basal activity of mTOR: the changes in cultured fibroblasts were recapitulated and widely distributed in central nervous system of the genetically coherent model of GM2 gangliosidoses in the Sandhoff strain mouse. Analysis of brain, spinal cord, brain stem and cerebellum revealed the same pathobiological abnormalities: diminished mTOR activity associated with reduced protein synthesis and increased autophagy. In this context it is notable that compensatory changes in mTOR activity have been reported in other Lysosomal storage disorders, including mucopolysaccharidosis type 2, Fabry disease, aspartylglucosaminuria and Pompe diseases in which impaired mTOR reactivation is associated with defective lysosome reformation [5,29]. Reduced basal mTOR activity has also been observed in diverse models of lysosomal diseases including Neuronal Ceroid Lipofuscinosis type 3 lymphoblastoid cells [30], in NPC1- and NPC2-knockdown endothelial cells [16], in a Drosophila model of mucolipidosis IV [31], and a human podocyte model of Fabry disease [32].

Autophagy is a constitutive but dynamically controlled process that is central to the maintenance of cellular homeostasis. Any disturbance of lysosomal function, as in the genetic disorders, Tay–Sachs and Sandhoff diseases will require compensatory adjustments to ensure, so far as possible, survival of the affected cell. These GM2 gangliosidoses, which preferentially affect the lyososomal recycling of membrane-derived sphingolipids abundant in neurons, provide a spectacular example of the rôle of autophagy in non-mitotic cells with a life-long dependence on mitochondrial energy generation: relentlessly progressive, these diseases cause widespread neuronal death. [33]. Our findings provide evidence for a mechanistic link between disrupted autophagy, increased permeabilsation of the lysosomal compartment and neuro-inflammatory changes [9,34].

We contend that the pathological mTOR signalling and consequential mitochondrial and lysosomal dysfunction that we report in GM2 gangliosidosis immediately suggest avenues for therapeutic exploration. Amino acid supplementation to restore mTOR activity has been investigated in Pompe disease: arginine and leucine were found to restore mTOR signalling and partially rescued the muscle disease due to gross failure of glycogen remodeling and with the accumulation of pathological autophagosomes in the sarcoplasm [5]. In our studies with fibroblasts obtained from affected patients, improvement of the cell phenotype was found to be specific for L-arginine, rather than L-leucine, and we attribute this specificity to the consistent genetic changes in a rate-limiting enzyme of L-arginine biosynthesis (arginosuccinate lyase) prompted by gene expression studies carried out in fibroblastsfrom patients with Tay–Sachs and Sandhoff diseases. Given that the patients have neurological dysphagia and in many cases feeding-tube placement, we at first considered that the reduced serum arginine concentrations might reflect a nutritional defect, especially in the most severely affected infants. However, our studies of fibroblasts were conducted in cells obtained after prolonged outgrowth culture of skin biopsy samples. In these, microarray analysis revealed a specific alteration related directly to a rate-limiting enzyme in arginine biosynthesis. The previous findings were corroborated by the reduced abundance of human argino-succinate synthetase protein in fibroblast extracts.

Tay–Sachs and Sandhoff diseases are paradigmatic examples of a large class of lysosomal diseases that are characterised by unremitting neurodegeneration. Unfortunately, treatments can only address symptoms and management is directed towards supporting critical functions that are progressively lost [35]. For these reasons, the conditions remain a focus of research-based principally on molecular therapies [7]. Our study considers a molecular approach that has not been explored in these diseases but is based on a fresh examination of the pathobiology of the disease. If adopted in practice, the approach is non-invasive and could be readily adapted to practical clinical care that includes a focus on nutrition. While it is an essential amino acid and natural dietary constituent, L-arginine has potent biological effects, for example on the formation of nitric oxide (for which it is the primary substrate) and in non-physiological doses has potential toxicity exerted by this and other mechanisms. Thus, any clinical use of L-arginine supplements would necessitate careful consideration, including approval from appropriate regulatory authorities and with the benefit of informed professional advice. However, modulation of mTOR with a physiological compound, L-arginine, if effective offers a largely non-invasive option for opportune exploration in GM2 gangliosidosis. After further clinical research, since it appears to have salutary effects on the cellular environment and attenuates the pathological release of cathepsins and other components of the lysosomal armoury, L-arginine might also be considered as an adjunct to definitive molecular therapies that directly address the genetic defect and are in development [6,7]. Here we conclude that our in vitro and in vivo studies call for further scientific exploration to support early-stage clinical studies. Furthermore, we corroborate the in vitro data after l-arginine treatment, and we can think about the possibility to design new therapeutic studies with arginine.

## 5. Innovation

Our work thoroughly and systematically shows that autophagy/lysosome/mTOR-associated molecules might be useful markers for monitoring the effects of potential therapeutic approaches beyond GM2 gangliosidosis. Thus, we provide evidence that L-arginine supplementation is a potential new treatment against lysosomal disorders that deserves being studied in further detail.

## Figures and Tables

**Figure 1 cells-10-03122-f001:**
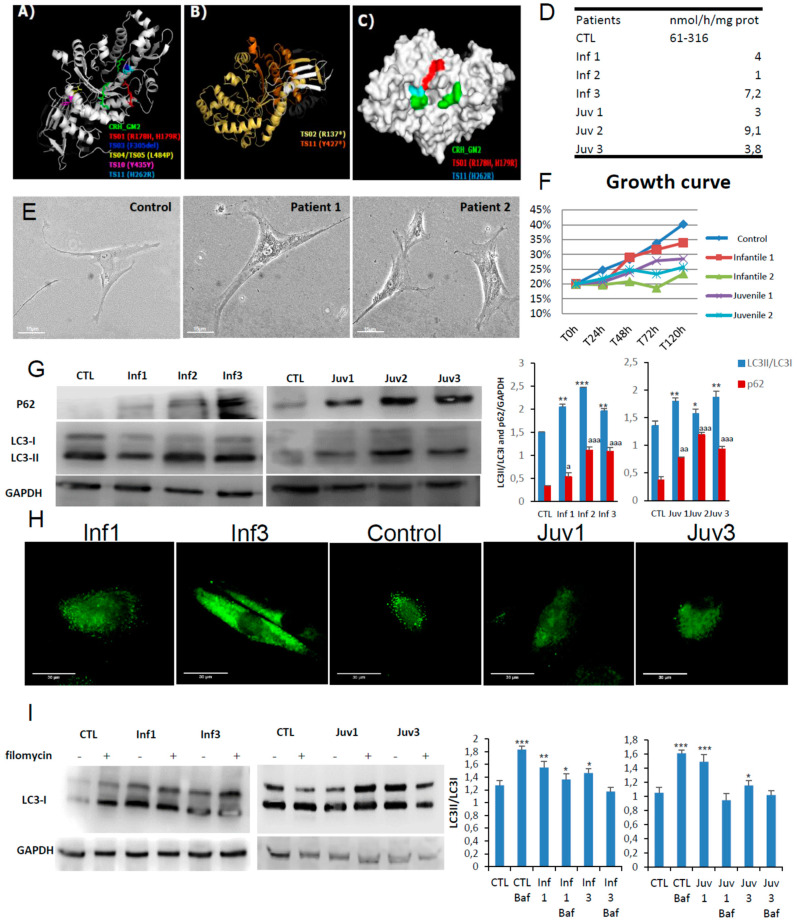
Model structures of HexA (PDB:2GJX) and HexB (PDB:1NOU) sub-unit proteins, highlighting the location of pathogenic mutations. Also shown autophagy in fibroblasts obtained in culture from patients with GM2 gangliosidosis (Tay–Sachs and Sandhoff diseases). (**A**). HexA point mutations: different colours depict amino acid substitutions identified in the cognate structures identified in different mutations studied. (**B**). Frameshift mutations in the alpha subunits found in two patients with Tay–Sachs disease are shown in yellow and orange; premature stop codons are marked by an asterisk. (**C**). The surface of hexosaminidase A with the critical active site region required for hydrolysis of GM2 ganglioside (CRH_GM2). The propeptide is shown in grey and the mature protein chain is depicted in white. (**D**). Enzymatic activity of HexA in fibroblast homogenates. (**E**). Morphological changes in fibroblasts from Tay–Sachs patients compared with control cells. (**F**). Cell growth determined in healthy and Tay–Sachs fibroblasts. (**G**). Expression of autophagy proteins in control and Tay–Sachs fibroblasts: LC3-I (top panels, top band), LC3-II (top panels, bottom band). (**H**). Immunofluorescence staining with anti-p62 antibody. (**I**). Impaired autophagic flux in Tay–Sachs fibroblasts. Determination of LC3-II in the presence and absence of bafilomycin A1 in control (CTL) and fibroblasts from Tay–Sachs patients; bafilomycin A1 was used at a final concentration of 100 nM with 12 h exposure. Total cellular extracts were analysed by immunoblotting with antibodies against LC3. The data are the mean ± SD for experiments conducted on two different control cell lines. Data represent the mean ± SD of three separate experiments. *** *p* < 0.001, ** *p* < 0.005, * *p* < 0.05 between cells from control subjects and patients with Tay–Sachs disease. ^a^
*p* < 0.05; ^aa^
*p* < 0.01; ^aaa^
*p* < 0.001.

**Figure 2 cells-10-03122-f002:**
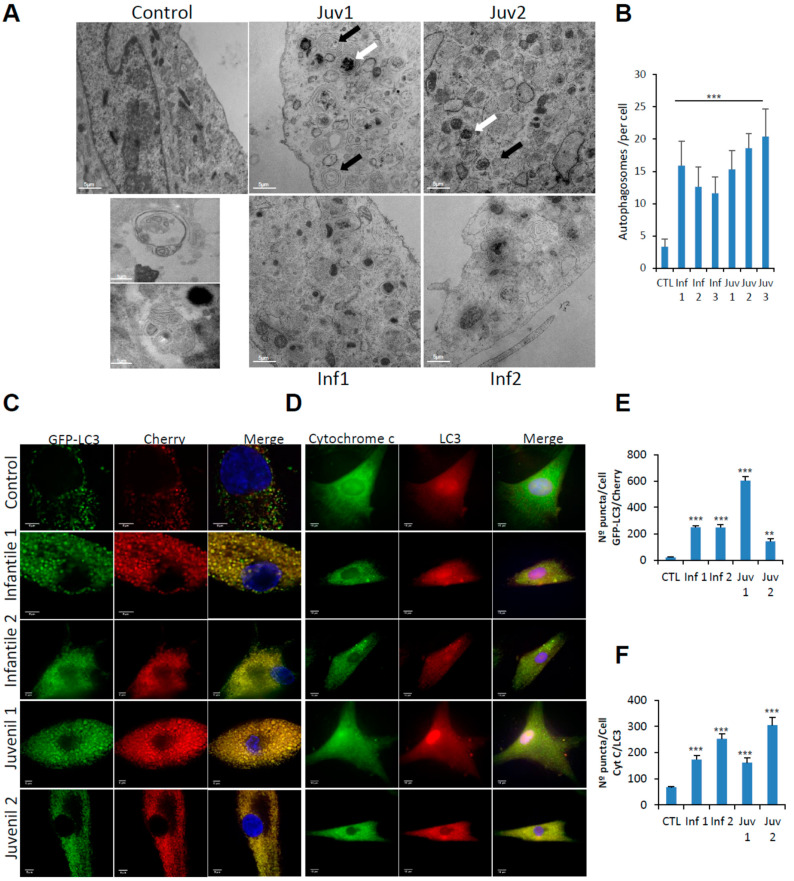
(**A**). Control fibroblasts and those from patients with Tay–Sachs disease showing typical ultrastructure with several distinct lamellar bodies (black arrows); white arrows indicate autophagosomes. Insets show multilamellar bodies (MLBs) and membrane-bound structures with cytoplasmic and cellular contents found in patient fibroblasts. Scale bar 10 µm (low magnification) and 2 µm (high magnification) (*n* = 20 cells per case). (**B**). Quantitative analysis of autophagosomes. (**C**,**E**). Representative image of fibroblasts after transfection of the dual-labelled mCherry-GFP-LC3 plasmid and quantification of autophagic puncta (see Methods). (**D**,**F**). Immunofluorescence of LC3 and cytochrome c in control and pathological cells and quantification of mitophagy puncta. Data represent the mean–SD of three separate experiments. *** *p* < 0.001, ** *p* < 0.005 between controls and Tay–Sachs patients.

**Figure 3 cells-10-03122-f003:**
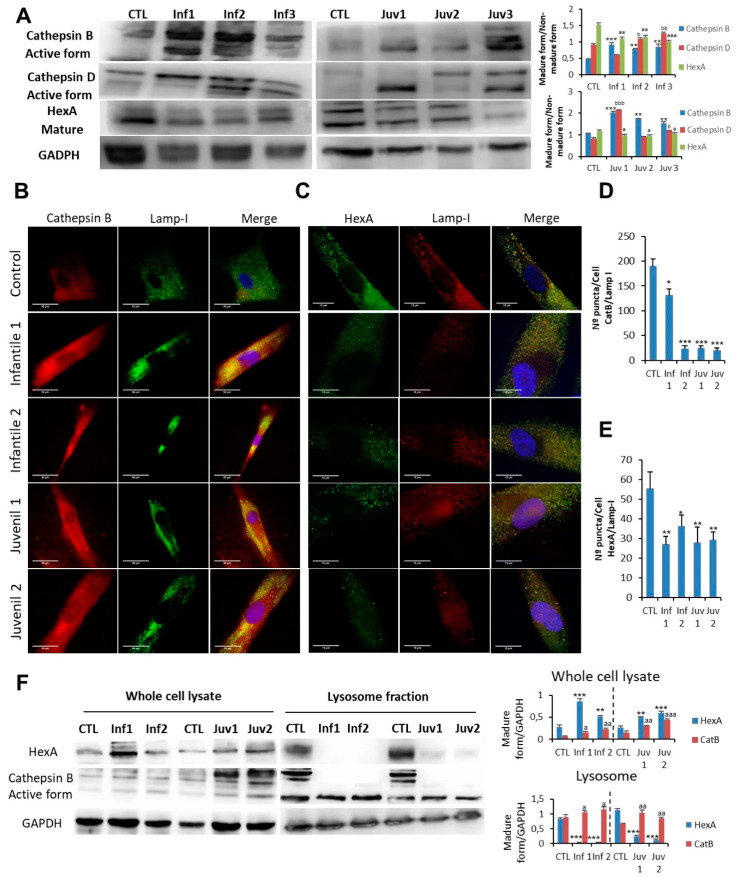
(**A**). Expression of CatB, CatD, and HexA protein were determined in control fibroblasts and those cultured from patients with Tay–Sachs disease. (**B**,**D**). Immunofluorescence of CatB in control and pathological cells and quantification. (**C**,**E**). Immunofluorescence of HexA in control and Tay–Sachs cells with signal quantification. Note that in Tay–Sachs fibroblasts CatB and HexA immunoreactivity is diffused throughout the cytosol. (**F**). Cellular fractionation with the isolation of cytosol and lysosomes and protein expression of CatB B and HexA. For control cells, results from two different control cell lines. Data represent the mean ± SD of three separate experiments. * *p* < 0.05; ** *p* < 0.01; *** *p* < 0.001 between control and patients with Tay–Sachs disease. ^a^ *p* < 0.05; ^aa^ *p* < 0.01; ^aaa^ *p* < 0.001; ^b^ *p* < 0.05; ^bb^ *p* < 0.01, ^bbb^
*p* < 0.001

**Figure 4 cells-10-03122-f004:**
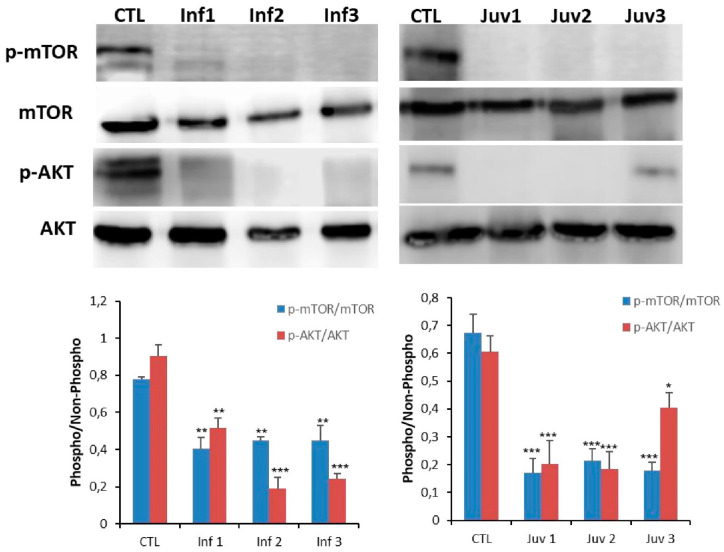
Expression of mTOR and AKT protein were determined in cultured control and Tay–Sachs disease fibroblasts. Data represent the mean ± SD of three separate experiments.* *p* < 0.05; ** *p* < 0.01; *** *p* < 0.001 between transfected and non-transfected cells.

**Figure 5 cells-10-03122-f005:**
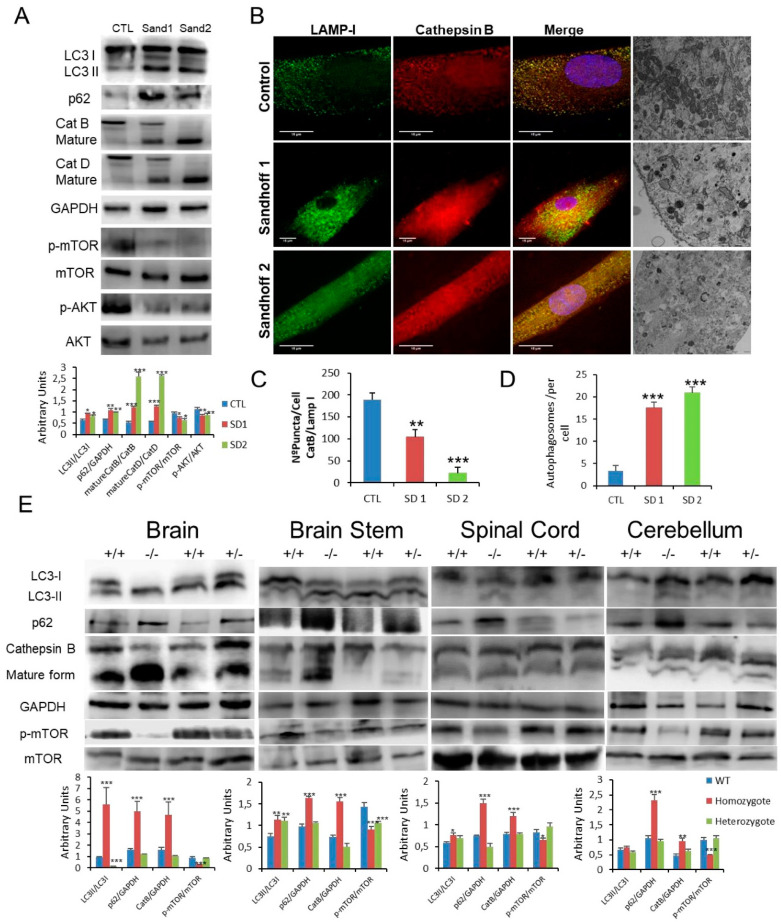
(**A**). Expression of LC3, p62, CatB, CatD, mTOR and AKT proteins determined in human control and Sandhoff disease fibroblasts. (**B**,**C**). Immunofluorescence of CatB in control and pathological cells with quantification in Sandhoff disease fibroblasts. (**B**,**D**). Characteristic ultrastructure with altered autophagosome abundance quantified in Sandhoff disease fibroblasts. (**E**). Expression of LC3, p62, CatB and mTOR proteins in the brain and spinal cord obtained from wild type and hexb −/− mutant mice with GM2 gangliosidosis (Sandhoff disease). Densitometry results are presented as means ± SEM, *n* = 10 mice. * *p* < 0.05; ** *p* < 0.01; *** *p* < 0.001 between control and diseased fibroblasts and wild type and hexB −/− mutant mice.

**Figure 6 cells-10-03122-f006:**
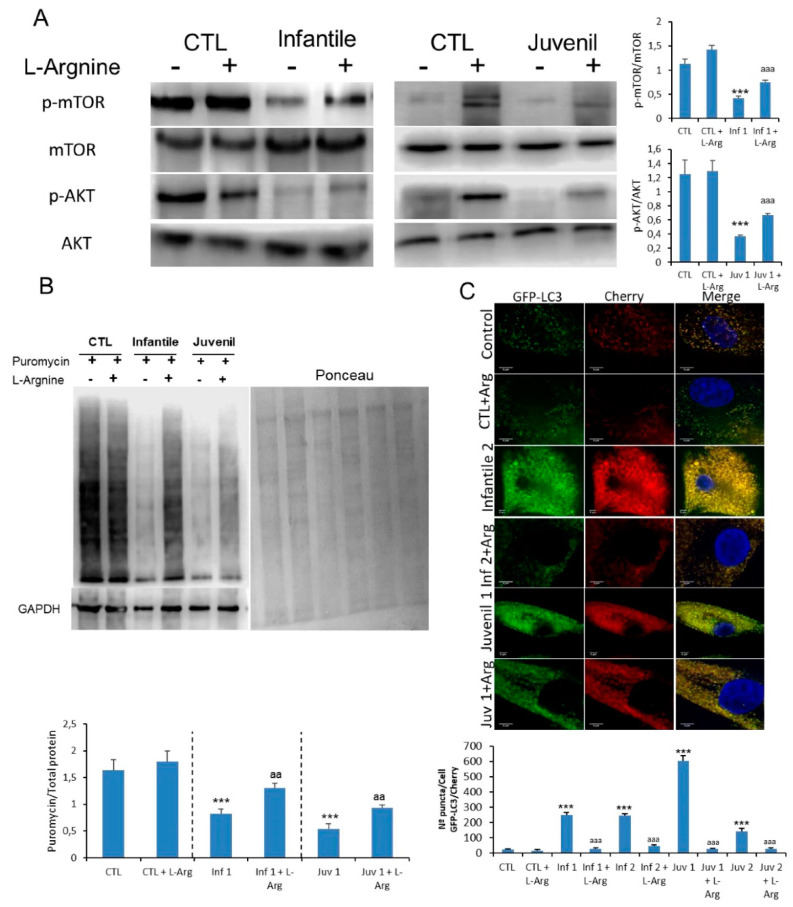
(**A**). Expression of mTOR and AKT determined in control and representative Tay–Sachs fibroblasts after L-arginine treatment. (**B**). Protein synthesis was quantified in extracts of control and Tay–Sachs fibroblasts treated with L-arginine using puromycin labeling followed by immunoblotting. (**C**). Representative image of Tay–Sachs treated fibroblasts after transfection of the mCherry-GFP-LC3 plasmid and quantification of autophagic puncta. For control cells, the data are the mean ± SD for experiments conducted on two different control cell lines. GAPDH was used as a loading control. Data represent the mean ± SD of three separate experiments. *** *p* < 0.001 between control and Tay–Sachs fibroblasts; ^aa^ *p* < 0.01; ^aaa^ *p* < 0.001 between non-treated and treated cells.

**Figure 7 cells-10-03122-f007:**
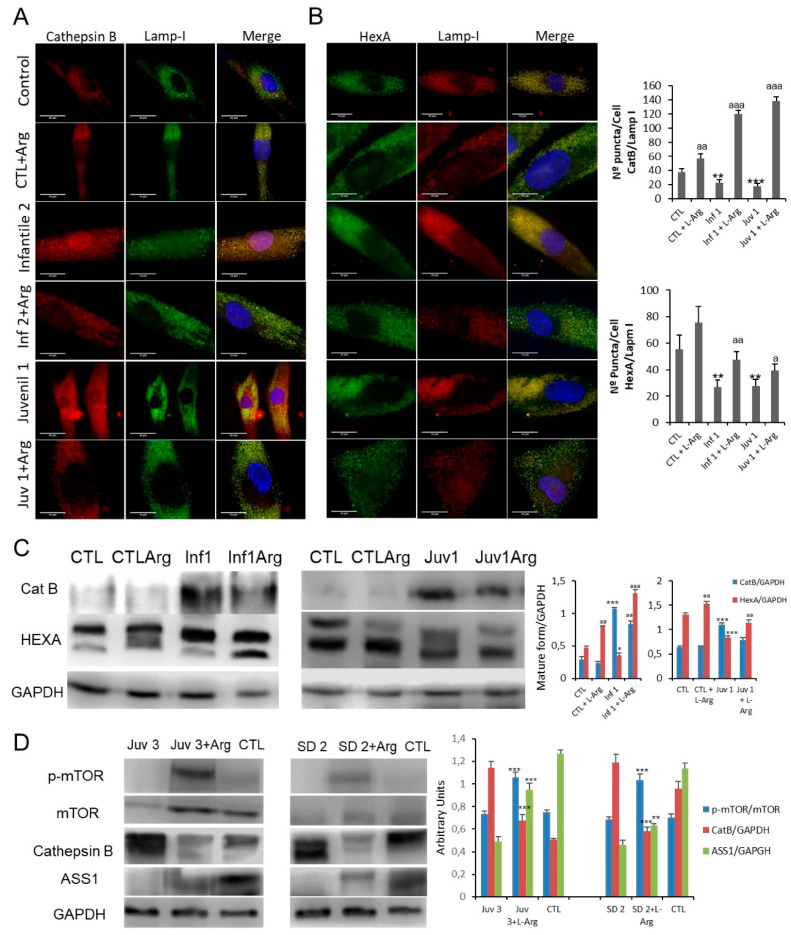
(**A**,**B**). Immunofluorescence of CatB and HexA in control and Sandhoff disease fibroblasts and quantification after L-arginine treatment. (**C**). Expression of CatB and HexA protein were determined in control and representative Tay–Sachs fibroblast cultures after L-arginine treatment in vivo. (**D**). Expression of mTOR, CatB, and ASS1 (arginosuccinate synthetase) proteins was determined in peripheral blood mononuclear cells obtained from a patient with juvenile Tay–Sachs disease and a patient with juvenile Sandoff disease after oral L-arginine treatment. Data represent the mean ± SD of three separate experiments.* *p* < 0.05; ** *p* < 0.01; *** *p* < 0.001 between control and Tay–Sachs patients; ^a^
*p* < 0.05; ^aa^ *p* < 0.01; ^aaa^ *p* < 0.001 between non-treated and treated cells.

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
