# Peer review of "L-Arginine Ameliorates Defective Autophagy in GM2 Gangliosidoses by mTOR Modulation"

_cells, 2021, doi:10.3390/cells10113122_

Round 1

Reviewer 1 Report

The manuscript of B. Castejon-Vega et al. entitled “L-arginine ameliorates defective autophagy in GM2 gangliosidoses by mTOR modulation” represents the data of research in which potential therapeutic targets involved in GM2 gangliosidoses, - Tay-Sachs and Sandhoff diseases, were studied. This pathology affects sphingolipid recycling in the nervous system reflecting accumulation of the primary substrate, GM2 ganglioside, in neuronal lysosomes due to impaired hydrolysis by β-hexosamidase A.

The main cause of GM2 gangliosidosis is a hereditary point mutations in HexA molecule, leading to loss of enzyme activity and resulting in multiple pathological consequences.

On studying the cellular pathogenesis of GM2 gangliosidosis, the authors found the following.

  1. Fibroblasts of Tay-Sachs patients had impaired morphology, growth rate and protein biosynthesis;
  2. Autophagy signaling is also impaired, as indicated by excessive accumulation of autophagy proteins, SQSTM1, LC3II and autophagosomes.
  3. Authors observed impaired digestion of mitochondria engulfed by lysosomes, a high level of cathepsin B and redistribution of HexA in cytosol, as well as nuclear translocation of transcription factor EB, suggesting impaired lysosomal permeability. Additionally, disturbance of mTOR/Akt signaling was also observed, in fibrobllasts as well as in nervous tissues (brain, brain stem, spinal cord, cerebellum)
  4. Transcriptomic analysis has shown changes in expression of genes related to mTOR signaling, autophagy and lysosomal processes.
  5. Authors have shown that the majority of cellular disturbances could be related to impairment of enzymes involved in arginine biosynthesis; moreover, arginine treatment recovers mTOR activity, cell growth, protein biosynthesis, and in part normalizes LC3II level, HexA and Cathepsin B intracellular distribution.

In general, this study looks logical, and conclusions are reliably supported by numerous and high-quality data.

Reviewer has no major remarks. This paper can be published with minor corrections.

Minor remarks.

  1. Line 144 – “Tris-ClH”
  2. Fig. 2A, - the arrows indicated in the legend to Fig 2A are absent in the figure itself.
  3. What does mean “madure form..” in ordinate to graphs in Fig 3F?!
  4. Line 321 – “…srelocalisation…”
  5. Line 331 - Impaired mTOR/autophagy and LMP are also associated with Sandhoff disease – What is “LMP”?
  6. Lines 324-326: “The down-regulated and phosphorylated forms of p-mTOR and p-AKT were increased in fibroblasts from patients with Tay-Sachs compared with control cells (Figure 4)” (see also Fig. 5E). – Are they really decreased or increased?
  7. Line 383 – “inlysosomal diseases”
  8. Line 402: To explore this phenomenon further, we investigated the kinase, PTEN, which canonically regulates…=PTEN is not kinase, rather phosphatase.

Author Response

The manuscript of B. Castejon-Vega et al. entitled “L-arginine ameliorates defective autophagy in GM2 gangliosidoses by mTOR modulation” represents the data of research in which potential therapeutic targets involved in GM2 gangliosidoses, - Tay-Sachs and Sandhoff diseases, were studied. This pathology affects sphingolipid recycling in the nervous system reflecting accumulation of the primary substrate, GM2 ganglioside, in neuronal lysosomes due to impaired hydrolysis by β-hexosamidase A.

The main cause of GM2 gangliosidosis is a hereditary point mutations in HexA molecule, leading to loss of enzyme activity and resulting in multiple pathological consequences.

On studying the cellular pathogenesis of GM2 gangliosidosis, the authors found the following.

  1. Fibroblasts of Tay-Sachs patients had impaired morphology, growth rate and protein biosynthesis;
  2. Autophagy signaling is also impaired, as indicated by excessive accumulation of autophagy proteins, SQSTM1, LC3II and autophagosomes.
  3. Authors observed impaired digestion of mitochondria engulfed by lysosomes, a high level of cathepsin B and redistribution of HexA in cytosol, as well as nuclear translocation of transcription factor EB, suggesting impaired lysosomal permeability. Additionally, disturbance of mTOR/Akt signaling was also observed, in fibrobllasts as well as in nervous tissues (brain, brain stem, spinal cord, cerebellum)
  4. Transcriptomic analysis has shown changes in expression of genes related to mTOR signaling, autophagy and lysosomal processes.
  5. Authors have shown that the majority of cellular disturbances could be related to impairment of enzymes involved in arginine biosynthesis; moreover, arginine treatment recovers mTOR activity, cell growth, protein biosynthesis, and in part normalizes LC3II level, HexA and Cathepsin B intracellular distribution.

In general, this study looks logical, and conclusions are reliably supported by numerous and high-quality data.

The authors want to thanks the positive and constructive comments to improve our manuscript.

Reviewer has no major remarks. This paper can be published with minor corrections.

Minor remarks.

  1. Line 144 – “Tris-ClH” The text has been corrected accordingly.
  2. Fig. 2A, - the arrows indicated in the legend to Fig 2A are absent in the figure itself. The figure has been corrected accordingly
  3. What does mean “madure form..” in ordinate to graphs in Fig 3F?!
  4. Line 321 – “…srelocalisation…” The text has been corrected accordingly
  5. Line 331 - Impaired mTOR/autophagy and LMP are also associated with Sandhoff disease – What is “LMP”? The text has been corrected accordingly
  6. Lines 324-326: “The down-regulated and phosphorylated forms of p-mTOR and p-AKT were increased in fibroblasts from patients with Tay-Sachs compared with control cells (Figure 4)” (see also Fig. 5E). – Are they really decreased or increased? The text has been corrected accordingly
  7. Line 383 – “inlysosomal diseases” The text has been corrected accordingly
  8. Line 402: To explore this phenomenon further, we investigated the kinase, PTEN, which canonically regulates…=PTEN is not kinase, rather phosphatase. Our apologies for this error, which has been corrected accordinly.

Reviewer 2 Report

In their study Castejón-Vega and colleagues show a still little-known aspect of Tay-Sachs disease, opening the way to a series of important questions.

Autophagy and mithophagy are widely described for many LSDs. Alterations of the autophagosome-lysosomal pathway are now documented in this work for Tay-Sachs disease.

The novelty of the study is evident. It is well done and further insights into the pathways studied can be postponed to a later work.

I highlight some points that can be strengthened:

  • the section of materials and methods is too generic for some methods (for example “Cathepsin B and HexA release”: you just say that CatB and HexA redistribution have been tested by immunofluorescence, without giving some info on staining, blocking, permeabilization, condition of Ab incubation).

  • improving of the immunofluorescence images must be indicated in the caption of respective figure (for example brightness +20% or contrast +5%)

  • Scale bar should be also indicated in EM images.

  • Some western blots shown need to be improved (figure 5E)

  • There are typos in the text that need to be corrected; for example: line 99 “skim”, line 332 “LMP”, line 337 table”1”.

Suggestion: Since you also show data on mitophagy in Figure 2 (D, F), you could use the images of EM in A to show and discuss mitochondrial status as well.

Author Response

In their study Castejón-Vega and colleagues show a still little-known aspect of Tay-Sachs disease, opening the way to a series of important questions.

Autophagy and mithophagy are widely described for many LSDs. Alterations of the autophagosome-lysosomal pathway are now documented in this work for Tay-Sachs disease.

The novelty of the study is evident. It is well done and further insights into the pathways studied can be postponed to a later work.

 I highlight some points that can be strengthened: 

  • the section of materials and methods is too generic for some methods (for example “Cathepsin B and HexA release”:you just say that CatB and HexA redistribution have been tested by immunofluorescence, without giving some info on staining, blocking, permeabilization, condition of Ab incubation).

The methodology has been improved accordingly

  • improving of the immunofluorescence images must be indicated in the caption of respective figure (for example brightness +20% or contrast +5%)

We want to thank to the reviewer for the comments to improve the quality of our manuscript. In this case, we have not used correction changes in our image, so all immunofluorescence have brightness 0% and contrast 0%.

  • Scale bar should be also indicated in EM images.

The figure has been corrected accordingly

  • Some western blots shown need to be improved (figure 5E)

Thank you for this comment to improve the quality of the figure. We know that the image could be improved however, with the circumstances on the investigators participant in this study and the pandemic, to repeat all this blots would not be realistically possible. The mouse model is from UK (Professor Cox) and now is imposible to send new mouse samples according to the Brexit from UK to Spain. Accordingly to this, we appreciate this comments and we could eliminate the blots image and include only the densitometric analysis in case the reviewer find this as a possibility.

  • There are typos in the text that need to be corrected; for example: line 99 “skim”, line 332 “LMP”, line 337 table”1”.

 The figure has been corrected accordingly

Suggestion: Since you also show data on mitophagy in Figure 2 (D, F), you could use the images of EM in A to show and discuss mitochondrial status as well.

We want to thank these comments from the reviewer, which help to improve our manuscript.

According to this comment, we are studying the role of mitochondria in Tay-sachs and Sandhoff diseases. However, we have not included these data in this manuscript because these data are included in other project with other research groups about the mitocondria these and other lysosomal disease. In that sense, we would prefer not include more data about this.

Reviewer 3 Report

In the article entitled ‘L-arginine ameliorates defective autophagy in GM2 gangliosidoses by mTOR modulation’ by Castejón-Vega et al., the authors have studied the effect of L-arginine on improving autophagy in the Lysosomal Storage Disorders (LSD) GM2 gangliosidosis Tay-Sachs and Sandhoff diseases. These diseases are characterized by the accumulation of GM2 ganglioside in neuronal cells. The authors have demonstrated that the mTOR signaling along with autophagic flux is defective in the fibroblasts derived from the GM2 gangliosidoses patients which can be rescued by supplementation with arginine. Previous studies have demonstrated the role of ERAD and apoptosis in the pathogenesis of Tay Sachs disease. However, the role of autophagy in the regulation of the GM2 gangliosidoses is an interesting aspect that has not been explored extensively and offers novelty to the study. The study is well designed, technically sound, and explores the role of autophagy in two different diseases with similar underlying disorders which increases the robustness of the study. The study is well suited for publication in the journal. However, there are a few concerns that should be addressed before the manuscript is taken into consideration by the journal for publication.

Major concerns:

  1. The process of autophagy and activation of the mTOR signaling pathway are anti-correlated. Interestingly, both the pathways have been shown to be similarly modulated (mTOR signaling decreased and autophagosome-lysosome fusion decreased at the same time). How do the authors justify this phenomenon?
  2. The images represented in figure 3B do not support the conclusions driven by the authors. The authors must include the colocalization coefficient parameter between CatB and LAMP1, and HexA and LAMP1 to support their conclusions. Also, the method for enumeration of the punctae should be included.
  3. The expression of LAMP1 in figure 3C is significantly lower compared to figure 3B. Could this be the underlying reason for the difference observed in the number of punctae?
  4. Is there a rationale to choose only arginine, leucine, and DL-acetyl leucine for the supplementation assays and not other amino acids?
  5. What is the effect of supplementation of arginine on oxidative phosphorylation/electron transport chain in the cells?
  6. As Tay Sach disease is characterized by accumulation of sphingolipids, what is the status of sphingolipid biosynthetic genes/sphingolipid content upon arginine supplementation?
  7. The authors must include a blot for LC3B and P62 upon arginine supplementation.

Minor concerns:

  1. Scale bar missing in figure 1e.
  2. The arrowheads are missing in figure 2a and must be included.
  3. The value for ‘n’ in figure 2b should be mentioned.
  4. The insets of TEM images in figure 2a should be labeled.
  5. The method for immunocytochemistry is missing along with the methodology for enumeration of LAMP1-positive punctae or LC3-positive structures.
  6. The color key in supplementary figure 13 must be included.

Author Response

In the article entitled ‘L-arginine ameliorates defective autophagy in GM2 gangliosidoses by mTOR modulation’ by Castejón-Vega et al., the authors have studied the effect of L-arginine on improving autophagy in the Lysosomal Storage Disorders (LSD) GM2 gangliosidosis Tay-Sachs and Sandhoff diseases. These diseases are characterized by the accumulation of GM2 ganglioside in neuronal cells. The authors have demonstrated that the mTOR signaling along with autophagic flux is defective in the fibroblasts derived from the GM2 gangliosidoses patients which can be rescued by supplementation with arginine. Previous studies have demonstrated the role of ERAD and apoptosis in the pathogenesis of Tay Sachs disease. However, the role of autophagy in the regulation of the GM2 gangliosidoses is an interesting aspect that has not been explored extensively and offers novelty to the study. The study is well designed, technically sound, and explores the role of autophagy in two different diseases with similar underlying disorders which increases the robustness of the study. The study is well suited for publication in the journal. However, there are a few concerns that should be addressed before the manuscript is taken into consideration by the journal for publication.

Major concerns:

  1. The process of autophagy and activation of the mTOR signaling pathway are anti-correlated. Interestingly, both the pathways have been shown to be similarly modulated (mTOR signaling decreased and autophagosome-lysosome fusion decreased at the same time). How do the authors justify this phenomenon?

Thank you for this comment about our data related to autophagy and mTOR. Our data show the consenced process between mTOR, decreased, with activation of autophagy, however, our observation show an excess of autophagy with impaired autophagosome-lysosome fusion. This probably could be associated to the excess of autophagosome accumulation in which lysosomes can not complete the fusión with all. This dysfunction has previously been proposed (PMID: 20974010).

  1. The images represented in figure 3B do not support the conclusions driven by the authors. The authors must include the colocalization coefficient parameter between CatB and LAMP1, and HexA and LAMP1 to support their conclusions. Also, the method for enumeration of the punctae should be included.

Pearson correlation by Deltavision system has been included in the results section and methodology has been improved accordingly.

  1. The expression of LAMP1 in figure 3C is significantly lower compared to figure 3B. Could this be the underlying reason for the difference observed in the number of punctae?

We agree that this comments is very interesting, however, in this case, we think both image groups are nor comparable because are two different experiments done in different momento with different conditions and different cells sown at different times.

  1. Is there a rationale to choose only arginine, leucine, and DL-acetyl leucine for the supplementation assays and not other amino acids?

This selection was chosed according to the previously described role of arginine and leucine to induce mTOR. Furthermore, DH-acetyl leucine was used by to be a modification of leucine which is being tested in Tay-Sachs patients for other groups. However, the alterations of the arginine biosynthesis pathway showed in the transcriptomic analysis, helped us to include more experiments using arginine supplementation.

  1. What is the effect of supplementation of arginine on oxidative phosphorylation/electron transport chain in the cells?

We want to thank these comments from the reviewer, which help to improve our manuscript. According to this comment, we are studying the role of mitochondria in Tay-sachs and Sandhoff diseases. However, we have not included these data in this manuscript because these data are included in other project with other research groups about the mitocondria these and other lysosomal disease. In that sense, we would prefer not include more data about this.

  1. As Tay Sach disease is characterized by accumulation of sphingolipids, what is the status of sphingolipid biosynthetic genes/sphingolipid content upon arginine supplementation?

Thank you for this comment to improve the quality of our manuscript. We agree that this observation would be very interesting to show the effect of arginine about the celular pathophysiology of Tay-Sachs, however, with the circumstances on the investigators participant in this study and the pandemic, to repeat this experiments would not be realistically possible. The skin fibroblasts from rare diseases usually have a limited lifespan and suffer negatively from cell freezing. So, in this moment, it would be imposible to isolate more skin fibroblasts from patients with the pandemic conditions.

  1. The authors must include a blot for LC3B and P62 upon arginine supplementation.

Again, and according to the previous response, in this moment, it would be imposible to isolate more skin fibroblasts from patients with the pandemic conditions.

Minor concerns:

  1. Scale bar missing in figure 1e.

This information has been included in the figure accorgingly

  1. The arrowheads are missing in figure 2a and must be included.

All the arrowheads have been included in the figure accordingly.

  1. The value for ‘n’ in figure 2b should be mentioned.

This info has been included accordingly

  1. The insets of TEM images in figure 2a should be labeled.

This info has been included in the text accordingly

  1. The method for immunocytochemistry is missing along with the methodology for enumeration of LAMP1-positive punctae or LC3-positive structures.

The methodology has been improved according to this reviewer and reviewer 2 comments

  1. The color key in supplementary figure 13 must be included.

The figure has been corrected accordingly

Round 2

Reviewer 3 Report

In the revision, the authors have addressed most of the minor concerns and provided explanations for the rest of them satisfactorily. It is understandable that repeating some experiments is difficult due to the limited availability of patient samples during pandemic. The manuscript may be accepted in the present form.